# Validation of CIP2A as a Biomarker of Subsequent Disease Progression and Treatment Failure in Chronic Myeloid Leukaemia

**DOI:** 10.3390/cancers13092155

**Published:** 2021-04-29

**Authors:** Richard E. Clark, Ammar A. Basabrain, Gemma M. Austin, Alison K. Holcroft, Sandra Loaiza, Jane F. Apperley, Christopher Law, Laura Scott, Alexandra D. Parry, Laura Bonnett, Claire M. Lucas

**Affiliations:** 1Department of Molecular and Clinical Cancer Medicine, University of Liverpool, Liverpool L69 3GA, UK; clarkre@liverpool.ac.uk (R.E.C.); a.a.basabrain@liverpool.ac.uk (A.A.B.); gemmajon2003@yahoo.co.uk (G.M.A.); alisonkholcroft@hotmail.co.uk (A.K.H.); l.scott1@liverpool.ac.uk (L.S.); 2John Goldman Centre for Cellular Therapy, Hammersmith Hospital, Imperial College Healthcare NHS Trust, London W12 0HS, UK; sandra.loaiza@nhs.net; 3Centre for Haematology, Imperial College London at Hammersmith Hospital, London W12 0HS, UK; j.apperley@imperial.ac.uk; 4Technology Directorate, University of Liverpool, Liverpool L69 3GA, UK; K.C.Law@liverpool.ac.uk; 5Chester Medical School, University of Chester, Bache Hall, Chester CH2 1BR, UK; alexparry@virginmedia.com; 6Department of Biostatistics, University of Liverpool, Liverpool L69 3GA, UK; L.J.Bonnett@liverpool.ac.uk

**Keywords:** CIP2A, CML, blast crisis, imatinib, dasatinib, SPIRIT2, disease progression

## Abstract

**Simple Summary:**

Tyrosine kinase inhibitor treatment has greatly improved the prognosis for many chronic myeloid leukaemia patients; however, disease progression is usually fatal for patients and remains a significant clinical challenge today. Using the UK SPIRIT 2 (STI571 Prospective International RandomIsed Trial 2) clinical trial, we have validated cancerous inhibitor of protein phosphatase 2A (PP2A) (CIP2A) as a diagnostic biomarker to identify patients at risk of disease progression and treatment failure. CIP2A is a simple diagnostic biomarker that may be a useful diagnostic tool in planning treatment strategies.

**Abstract:**

Background: It would be clinically useful to prospectively identify the risk of disease progression in chronic myeloid leukaemia (CML). Overexpression of cancerous inhibitor of protein phosphatase 2A (PP2A) (CIP2A) protein is an adverse prognostic indicator in many cancers. Methods: We examined CIP2A protein levels in diagnostic samples from the SPIRIT2 trial in 172 unselected patients, of whom 90 received imatinib and 82 dasatinib as first-line treatment. Results: High CIP2A levels correlated with inferior progression-free survival (*p* = 0.04) and with worse freedom from progression (*p* = 0.03), and these effects were confined to dasatinib recipients. High CIP2A levels were associated with a six-fold higher five-year treatment failure rate than low CIP2A levels (41% vs. 7.5%; *p* = 0.0002), in both imatinib (45% vs. 11%; *p* = 0.02) and dasatinib recipients (36% vs. 4%; *p* = 0.007). Imatinib recipients with low CIP2A levels had a greater risk of treatment failure (*p* = 0.0008). CIP2A levels were independent of Sokal, Hasford, EUTOS (EUropean Treatment and Outcome Study), or EUTOS long-term survival scores (ELTS) or the presence of major route cytogenetic abnormalities. No association was seen between CIP2A levels and time to molecular response or the levels of the CIP2A-related proteins PP2A, SET, SET binding protein 1 (SETBP1), or AKT. Conclusions: These data confirm that high diagnostic CIP2A levels correlate with subsequent disease progression and treatment failure. CIP2A is a simple diagnostic biomarker that may be useful in planning treatment strategies.

## 1. Introduction

Tyrosine kinase inhibitors (TKIs) have transformed the clinical landscape for patients with chronic-phase chronic myeloid leukaemia (CML), with life expectancy essentially normal for about 90% of patients. However, TKIs do not offer a cure, as they do not eradicate leukaemic stem cells (LSC) [1,2,3]. Furthermore, a proportion of patients will progress to blast crisis, with a median survival of approximately 10 months [4]. Disease progression to accelerated phase is defined by the European Leukemia Net (ELN) as a blast count >15% and blast crisis is >30% [5,6]. Various scoring systems have been developed to prospectively identify patients at high risk of transformation [7,8,9], but Italian data on the long-term outcome of imatinib treatment suggest that >70% of patients identified as high risk by Sokal or other scores may remain well at 7 years [10] precluding the use of these scoring systems to influence treatment decisions. More recently, the EUTOS long-term survival score (ELTS) has been developed to look at the risk of disease progression to the advanced phase, but again even patients in the high-risk ELTS group had an 8-year incidence of death from CML of only 11% [11]. Similarly, the German/Swiss CML IV study has identified that certain additional chromosomal abnormalities (ACA) beyond a single Philadelphia (Ph) translocation may confer a six-fold increased risk of disease progression [12], but ACA are only present in ~3% of patients, and also these findings have yet to be confirmed in other studies. There is, therefore, a need for a reliable prospective biomarker of disease progression to blast crisis.

Protein phosphatase 2A (PP2A) is a phosphatase that is important in opposing the overactive kinases that typify many malignancies; it is thus a tumour suppressor. PP2A inactivation is an essential requirement for transforming human cells [13]. PP2A is regulated by numerous inhibitor proteins such as a Cancerous Inhibitor of PP2A (CIP2A) [14], SET [15], and SET binding protein 1 (SETBP1) [16], and expression of these inhibitor proteins can vary between malignancies and even within malignancy sub-types. For example, acute myeloid leukaemia (AML) patients with normal karyotype have their PP2A predominately inhibited by CIP2A, while patients with an adverse karyotype have their PP2A inhibited by SETBP1 [17]. Both SETBP1 and CIP2A inhibit PP2A, resulting in phosphorylation of the serine/threonine kinase AKT (also known as protein kinase B) at its S473 residue [17], and this leads to uncontrolled cell proliferation and resistance to apoptosis, both hallmarks of oncogenic transformation [18].

We reported in 2011 that at initial diagnosis of chronic-phase CML, imatinib recipients with high levels of CIP2A protein had a high actuarial risk of progression to blast crisis, and this was not associated with *BCR-ABL1* kinase domain mutations [19]. Over 150 clinical papers have subsequently confirmed that high CIP2A levels are associated with adverse histology and/or poor response to treatment in over 20 different cancer histologies [20,21,22,23,24,25,26], including AML [17]. Our group subsequently showed that patients with high CIP2A levels who received front-line treatment with the second generation (2G) TKIs dasatinib or nilotinib did not have this high progression risk, suggesting that CIP2A might be a useful biomarker to identify a group of patients who might be better treated with a 2G TKI [27]. However, our studies were limited to 74 patients in whom disease progression was overrepresented but confined to imatinib recipients; indeed, no samples were available at our institution from front-line dasatinib or nilotinib recipients that underwent progression. It is therefore important to validate our findings in an independent and larger cohort more representative of everyday CML practice. Therefore, we investigated the prospective value of assessing CIP2A and related proteins at diagnosis in a subset of 172 patients enrolled in the United Kingdom SPIRIT2 trial, a phase 3 study comparing imatinib and dasatinib for first-line treatment in newly diagnosed chronic-phase CML patients. The principal findings from SPIRIT2 have been presented [28].

## 2. Materials and Methods

### 2.1. Patients

In the SPIRIT2 trial, 814 newly diagnosed chronic-phase patients were randomly allocated 1:1 to either imatinib 400 mg or dasatinib 100 mg each once daily. Follow-up was monthly for 3 months, 3-monthly until 12 months, then 6-monthly. Patients were followed until the sooner of 5 years or a change of therapy due to either intolerance or resistance [28]. At each visit, a centralised molecular assessment of the *BCR-ABL1/ABL1* ratio was carried out using International Standardisation (IS) at the Molecular Pathology Laboratory at the Imperial College Healthcare NHS Trust (ICHNT), London. Results with less than 10,000 *ABL1* control transcripts were rejected [29].

The vast majority of SPIRIT2 entrants gave informed consent to donate samples to the SPIRIT2 biobank housed at ICHNT, in addition to the consent required to enter the trial. The present project was approved by the National Cancer Research Institute CML subgroup, who have ownership of this biobank, and ethical approval was given by the Liverpool East Committee of the UK. National Research Ethics Committee. To maximise follow-up yet conserve resources, the study focused on the first available 200 (25%) diagnostic samples from the SPIRIT2 entrants plus all subsequent patients who progressed. This resulted in a total of 172 samples available/suitable for study, which included 18 of the 23 patients in the entire study who progressed to the advanced phase.

### 2.2. Sample Collection and Preparation

Peripheral blood mononuclear cells were separated from diagnostic samples by density-dependent centrifugation (Lymphoprep Axis-Shield, Cambridge, UK), washed in RPMI 1640 (BioSera, Nuaille, France), and resuspended in 10% dimethylsulfoxide (DMSO)/10% fetal calf serum (FCS) (BioSera)/RPMI at 4 °C. Cells were then cryopreserved. Samples were thawed in RPMI containing 10% FCS and 1% L-glutamine using the dropwise method.

### 2.3. Measurement of CIP2A and Associated Proteins

The flow cytometry methodology has been previously described [17,19,27] and was used for the detection of CIP2A, SET, SETBP1, AKT, and AKT^S473^. The following antibodies were used: CIP2A, SET, SETBP1, AKT, and AKT^S473^ (all from Santa Cruz Biotechnology, California, USA), anti-mouse and anti-rabbit Alex Fluor 488 (Invitrogen, Paisley, UK), and CD45perCP (BD Bioscience, Oxford, UK). Samples were analysed by flow cytometry using a FACSCanto II machine (BD Bioscience, Oxford, UK), and the resultant data were analysed by FACS Diva software, version 8.0.1.

### 2.4. Definitions of Outcome Endpoints

Overall survival (OS) was defined as the time from trial entry to death from any cause. Progression-free survival (PFS) was defined as the time from trial entry to disease progression to advanced phase or death from any cause, whichever were the earlier. Freedom from progression (FFP) was defined as the time from trial entry to progression alone. Time to treatment failure was defined as the time from trial entry to a change in the allocated therapy because of resistance.

### 2.5. Statistical Analysis

The level of CIP2A was assigned as either high or low according to the maximally selected rank statistic, which is an outcome-oriented method providing a value of a cut point that corresponds to the most significant relationship with the outcome. In Kaplan–Meier plots, *p*-values were determined using the log-rank (Mantel–Cox) test; where significant *p*-values are shown. All analyses were undertaken in R 3.5.0 and GraphPad prism v8.1.

## 3. Results

Patient characteristics for all 172 patients studied are shown in Table 1. Briefly, 90 patients received imatinib, while 82 received dasatinib. Eighteen patients progressed to blast crisis (9 imatinib- and 9 dasatinib-treated patients). Overall, the median age was 52 years, similar in each treatment arm, though 115 (66.9%) of the 172 patients were male, with 61.1% and 73.3% in the imatinib and dasatinib arms, respectively.

### 3.1. CIP2A and Established Scoring Sytems

We first examined how the established scoring systems correlated to progression and to levels of CIP2A and its related proteins. Recording the components of the various scoring systems was not mandatory at trial entry; as a result, 39% (Sokal), 45% (Hasford), 10% (EUTOS), and 39% (ELTS) of the present 172 patients could not be allocated a score. There was a non-significant trend for worse progression-free survival in patients with a high Sokal score (Figure 1a) and a similar trend for patients with high Sokal scores to have higher CIP2A protein levels (Figure 1b). The Hasford and EUTOS scores did not predict progression-free survival (consistent with the findings of both the Italian and German studies) [8,10] or correlate with CIP2A levels (Figure 1c–f). However, the ELTS score correlated with progression-free survival (Figure 1g; *p* = 0.01), consistent with a recent report [30], though no correlation was seen between it and CIP2A levels (Figure 1h). No association was found between diagnostic CIP2A levels and the presence of trisomy 8 or 19, a second Ph translocation, or isochromosome 17 (Figure 1i). CIP2A protein levels are therefore independent of the scoring systems and abnormal cytogenetics.

### 3.2. High CIP2A Is Associated with an Inferior Progression-Free Survival

The overall survival for these 172 patients is given in Appendix A, according to the patient’s CIP2A status and stratified by TKI received. No correlation is seen between the CIP2A level and overall survival.

Figure 2a shows that a high CIP2A level is associated with inferior progression-free survival (*p* = 0.04). Figure 2b,c shows the progression-free survival for the imatinib and dasatinib recipients, respectively, and suggests that the trend toward inferior progression-free survival with high CIP2A mostly derives from the dasatinib recipients, though numbers become too small to achieve statistical significance. However, the contribution of progression to progression-free survival is dominated by patients who died from causes unrelated to CML. In the present 172 patients, 27 deaths were observed, of which only 14 (51%) were CML-related. Therefore, in assessing the effect of CIP2A on progression, freedom from progression (FFP) may be more informative. This is shown in Figure 3, set out in the same way as Figure 2. As for progression-free survival, high CIP2A levels are associated with inferior freedom from progression (Figure 3a; *p* = 0.03, Fisher’s exact test), and again this is mostly due to the dasatinib recipients (Figure 3c; *p* = 0.003, Fisher’s exact test); no difference in freedom from progression according to CIP2A status is seen in the imatinib recipients (Figure 3b).

Overall, the risk of disease progression for patients with low and high CIP2A levels was 5% and 15%, respectively (Figure 3a), but this difference was particularly marked for dasatinib recipients (Figure 3c; 2% and 23%, respectively; *p* = 0.004), while not differing for imatinib recipients (Figure 3b; 9.5% and 10%, respectively). One dasatinib-treated patient with low CIP2A progressed; this patient had a high Sokal score and an intermediate ELTS score and frequent treatment interruptions due to recurrent pleural effusions. Four out of 42 imatinib recipients with low CIP2A progressed, and all four presented with intermediate (2 patients) or high (2 patients) Sokal and ELTS scores, and 3 of the 4 had ACA at diagnosis (either trisomy 8, a second Ph, or isochromosome 17). The adverse cytogenetics and treatment interruption may have contributed to progression in these patients despite their low diagnostic CIP2A level.

### 3.3. High CIP2A Is Associated with Treatment Failure

We next investigated if the diagnostic CIP2A level was associated with treatment failure. Figure 4a shows that patients with high CIP2A levels were more likely to fail treatment (*p* = 0.001). When patients were stratified by TKI treatment, there was a trend for imatinib-treated patients with high CIP2A levels to have a higher failure rate than those with low levels. For dasatinib-treated patients, high CIP2A levels conferred a significantly higher risk of treatment failure (Figure 4c; *p* = 0.003). Patients with low CIP2A levels and treated with imatinib had an inferior treatment failure rate than dasatinib-treated patients (Figure 4d; *p* = 0.0008). Patients with high CIP2A levels had a significant risk of failing treatment (*p* = 0.001); when we compared imatinib and dasatinib treatment for patients with high CIP2A levels, imatinib-treated patients had a trend for higher risk of treatment failure (Figure 4e).

Patients with a high CIP2A level at diagnosis had a 41% chance of treatment failure by 5 years, compared to 7.5% for patients with low CIP2A levels (*p* = 0.002, Fisher’s exact test). For imatinib-treated patients, the risk of treatment failure was 45% and 11% for the high and low CIP2A groups, respectively. (*p* = 0.02, Fisher’s exact test). For dasatinib-treated patients, patients with high CIP2A levels had a 36% chance of treatment failure compared to 4% for those with low CIP2A levels (*p* = 0.007, Fisher’s exact test). High CIP2A levels at diagnosis thus predict a significant risk of treatment failure irrespective of TKI treatment.

### 3.4. Time to Molecular Response

Imatinib recipients with a high CIP2A level had a significantly worse early molecular response rate (defined as a *BCR-ABL1/ABL1^IS^* ratio of ≤10% at 3 months) (*p* = 0.04; data not shown). The time to molecular response 3 (MR3) (*BCR-ABL1/ABL1^IS^* ratio of ≤0.1%) and time to molecular response 4.5 (MR4.5) (*BCR-ABL1/ABL1^IS^* ratio of ≤0.0032%, in the presence of at least 31,623 control *ABL1* transcripts) are shown in Figure 5. The diagnostic CIP2A level, therefore, did not correlate with time to molecular response, either overall or for imatinib or dasatinib recipients alone.

### 3.5. Prognostic Value of CIP2A/PP2A Related Proteins at Diagnosis

CIP2A exerts its effects through association with a number of network-related proteins [31,32,33]. These include PP2A itself [14], SET [34], SETBP1 [16,17], and S473 phosphorylation of AKT [17]. We investigated the prognostic value of these CIP2A/PP2A-related proteins at diagnosis. Patients with a high SET protein level had inferior freedom from progression (*p* = 0.01, data not shown) and a trend towards inferior progression-free survival, compared to patients with low diagnostic SET levels. In AML, high expression of SETBP1 at diagnosis is a marker of poor survival [14], and we have recently shown that high diagnostic levels of AKT^S473^ at diagnosis are also associated with poor outcome [17]. Here the diagnostic level of SETBP1 did not offer any prognostic value, and although high levels of total AKT protein were associated with an inferior time to molecular response 2 (MR2) for dasatinib-treated patients (*p* = 0.05), no prognostic significance for AKT^S473^ was found. We therefore conclude that the levels of these CIP2A-related proteins offer no advantages over CIP2A itself in predicting outcome in CML.

## 4. Discussion

Here we examine diagnostic CIP2A levels in a large clinical trial and demonstrate that high CIP2A levels are associated with a significantly higher probability of disease progression. However, in contrast to our previous studies (where we were unable to comment on dasatinib recipients as none progressed), this relationship between CIP2A on progression was not seen in imatinib recipients. This may be because a higher proportion of the present imatinib recipients switched treatment because of resistance than dasatinib recipients, and the rate of stem cell transplantation was five-fold higher for imatinib recipients than dasatinib recipients in the trial overall [28], and these treatment alterations may have prevented disease progression. This requires further study.

The present data also indicate that high CIP2A is associated with a higher chance of treatment failure for both imatinib and dasatinib recipients. Higher levels of CIP2A are associated with a higher degree of PP2A inhibition and are known to confer treatment resistance in a wide range of other tumours (reviewed in [35]). Little is known about CIP2A’s mechanism of action. A direct interaction for CIP2A and PP2A has been described, showing CIP2A binds at least two PP2A regulatory subunits, β56γ and β56α [36]. The structure of the N-terminal region (residues 1–560) of CIP2A has been determined. This region is important in facilitating PP2A binding, as it contains the homodimer contacts spanning residues 507–559 that enhance CIP2A’s binding to PP2A subunits. The minimum region required for CIP2A binding to β56α and β56γ has been identified as residues 159–245 [36]; however, a structure containing both CIP2A and PP2A components has yet to be identified. Whether these interactions fulfil all CIP2A binding is unknown as, despite efforts by us and others, the full-length protein is unstable and has not yielded any crystals for X-ray crystallographic structural study.

A plausible explanation for the adverse effects of CIP2A on treatment outcome may be that patients whose disease is set to have a particularly high level of PP2A inhibition are already destined to respond less well to treatment and are at greater risk of the additional genetic events required for disease progression. Junttila et al. [37] first described that CIP2A associates with and stabilises MYC in Hela cells, increasing MYC’s half-life and its activation (indicated by phosphorylation on serine 62) as well as promoting MYC localisation to nuclear lamins in cancer cell lines [37,38,39,40]. Furthermore, in the CML cell line K562 [19,41] and primary CML patient cells [27,42], we have confirmed this association. High levels of MYC are known to promote genetic instability and aneuploidy and thus may contribute to disease progression. We have previously shown that MYC inhibition can inhibit CIP2A via a positive feedback loop [27,42]. This interaction is direct and involves the N-terminal residues 1–262 of MYC. The mechanism for CIP2A’s regulation on MYC is likely through its inhibitory effect on PP2A-mediated MYC dephosphorylation (which inactivates MYC and decreases its stability). However, CIP2A may also have additional actions. In separate work, we have recent evidence that one mechanism of CIP2A action is to alter the balance of pro- and anti-apoptotic proteins in favour of creating an anti-apoptotic phenotype [43]. Furthermore, CIP2A augments oxidative phosphorylation and decreases reliance on glycolysis, directly interacting with a number of energy metabolism proteins in a manner suggesting that these metabolic effects may be mediated through 5′ adenosine monophosphate-activated protein kinase (AMPK), since modifying AMPK activity mimics the effects of CIP2A on energy metabolism [41]. These metabolic effects appear to be a novel action of CIP2A in malignancy.

## 5. Conclusions

In summary, this study on a large, unselected trial cohort is in line with our earlier observations on selected local patients [19,27] and suggests that diagnostic CIP2A protein levels could be used at diagnosis as a potential biomarker for predicting progression and treatment failure.

## Figures and Tables

**Figure 1 cancers-13-02155-f001:**
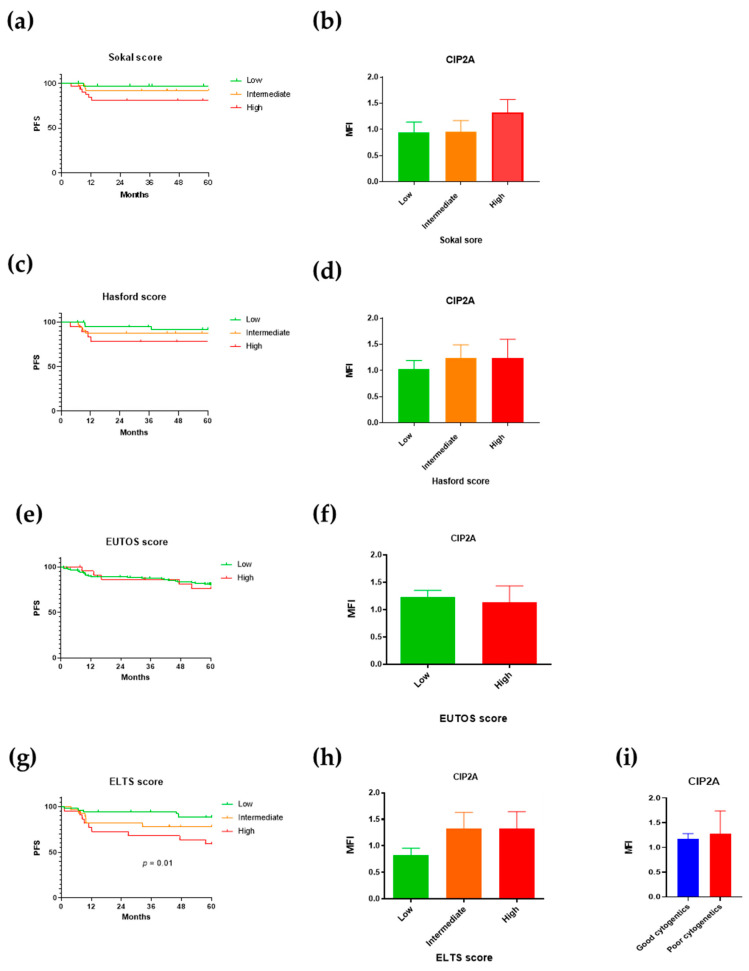
CIP2A (cancerous inhibitor of protein phosphatase 2A) protein level is independent of scoring systems and abnormal cytogenetics. Progression-free survival (PFS) and diagnostic CIP2A MNC protein levels stratified by Sokal score (**a**,**b**), Hasford score (**c**,**d**), EUTOS (European Treatment and Outcome Study) score (**e**,**f**), ELTS (EUTOS long-term survival scores) (**g**,**h**), and abnormal cytogenetics (**i**).

**Figure 2 cancers-13-02155-f002:**
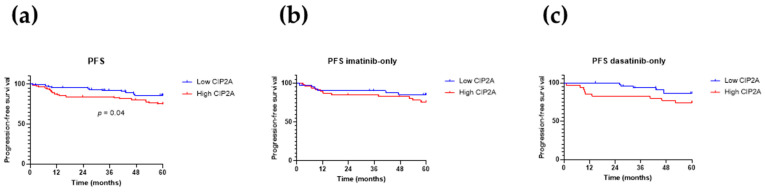
Progression-free survival (PFS). Kaplan–Meier curves for progression-free survival, stratified by diagnostic CIP2A level for (**a**) all 172 patients, (**b**) imatinib recipients, and (**c**) dasatinib recipients. *p*-values were determined using the log-rank (Mantel–Cox) test; *p*-values are shown where significant.

**Figure 3 cancers-13-02155-f003:**
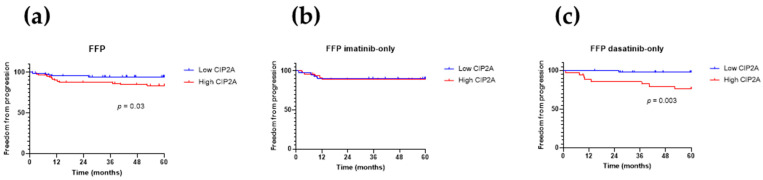
Freedom from progression (FFP). Kaplan–Meier curves for freedom from progression, stratified by diagnostic CIP2A level for (**a**) all 172 patients, (**b**) imatinib recipients, and (**c**) dasatinib recipients. *p*-values were determined using the log-rank (Mantel–Cox) test; *p*-values are shown where significant.

**Figure 4 cancers-13-02155-f004:**
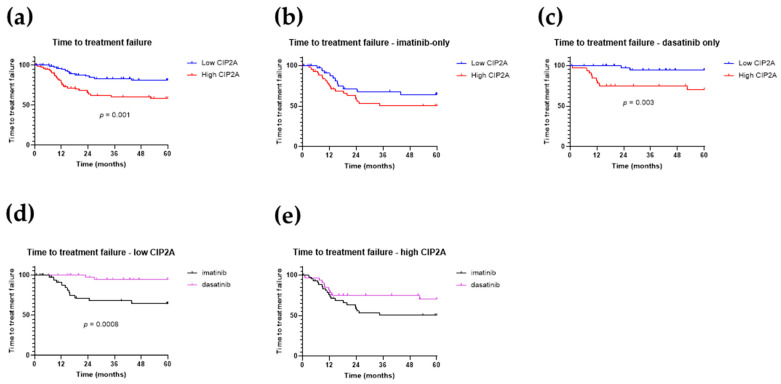
Time to treatment failure. Kaplan–Meier curves for time to treatment failure, stratified by diagnostic CIP2A level for (**a**) all 172 patients, (**b**) imatinib recipients, and (**c**) dasatinib recipients, (**d**) according to treatment received for patients with low CIP2A levels only, and (**e**) according to treatment received for patients with low CIP2A levels only. *p*-values were determined using log-rank (Mantel–Cox) test; *p*-values are shown where significant.

**Figure 5 cancers-13-02155-f005:**
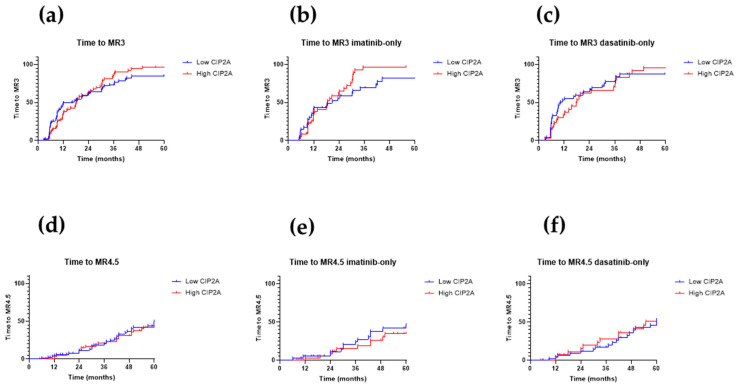
Time to molecular response— molecular response 3 (MR3) and molecular response 4.5 (MR4.5). Panels **(a–c)**: Kaplan–Meier curves for time to MR3, stratified by diagnostic CIP2A level for (**a**) all 172 patients, (**b**) imatinib recipients, and (**c**) dasatinib recipients. Panels (**d**–**f**): Time to MR4.5 stratified by diagnostic CIP2A level for (**d**) all 172 patients, (**e**) imatinib recipients only, and (**f**) dasatinib recipients only. *p*-values were determined using the log-rank (Mantel–Cox) test; *p*-values are shown where significant.

**Table 1 cancers-13-02155-t001:** Patient characteristics table.

Catergory	Imatinib (90)	Dasatinib (82)	Total
Median age (range)	52 (20–87)	55 (18–81)	52 (18–87)
Sex			
Male	55	60	115
Female	35	22	57
Sokal Score			
Low	20	14	34
Intermediate	16	22	38
High	18	14	32
N/A	36	32	68
Hasford Score			
Low	23	18	41
Intermediate	15	18	33
High	12	7	19
N/A	40	39	79
EUTOS			
Low	73	58	131
High	11	12	23
N/A	6	12	18
ELTS			
Low	28	26	54
Intermediate	14	14	28
High	12	10	22
N/A	36	32	68

N/A = not available; other abbreviations as defined in the text.

## Data Availability

The data presented in this study are available within the article or Appendix A.

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
