# Peer review of "Validation of CIP2A as a Biomarker of Subsequent Disease Progression and Treatment Failure in Chronic Myeloid Leukaemia"

_cancers, 2021, doi:10.3390/cancers13092155_

Round 1

Reviewer 1 Report

Authors described about the association between CIP2A level and the prognosis of patients with CML treated with second generation TKI. Some issues is concerned. 1. CIP2A protein level is assessed by FACS. Sample is peripheral blood, but what is the fraction? CD34+CD38-CD26+ CML stem cell? It is unclear. 2. In Figure 2A shows that a high CIP2A level is associated with inferior PFS. Whereas, in figure 5, the diagnostic CIP2A level therefore did not correlate with time to MMR or MR4.5. CML progression is generally defined as BCR-ABL1 mRNA transcript level by PCR. Why the differences existed between PFS and achievement of MMR or MR4.5? 3. In discussion part, authors explained the CIP2A immunoprecipitation experiments in Hela cells, CIP2A associates with and stabilises MYC. Is there any paper using CML cell lines? K562, KBM5 etc.

Reviewer 2 Report

Protein phosphatase 2A (PP2A) is a trimeric holoenzyme, which plays an integral role in regulating many cancer signaling pathways. Therefore, PP2A appears to be involved in the development of cancer. Indeed, PP2A has been indicated to exert tumor suppressive function that loss-of-function of PP2A contributes to tumor development and progression. Functional inactivation of PP2A tumor suppressor activity occurs in CML with myeloid blast crisis via the effect of BCR-ABL on SET expression. Re-establishing functional inactivation of PP2A is essential for BCR-ABL leukemogenesis in vivo. Cancerous inhibitor of PP2A (CIP2A) is a recently described inhibitor of PP2A in breast and gastric cancer. It interacts directly with the oncogenic transcription factor c-Myc, inhibits PP2A activity toward c-Myc serine 62 (S62), and thereby prevents the proteolytic degradation of c-Myc. Except for the function in c-Myc stabilization, CIP2A can also promote the growth of anchorage-independent cell and tumor formation in vivo. CIP2A has been reported to be overexpressed in multiple human solid tumors. In CML, it has been shown that high levels of CIP2A at diagnosis is a prospective biomarker for future progression to blast crisis. High levels of CIP2A in other malignancies have also been reported to confer a poor prognosis. In CML, high CIP2A levels lead to the stabilization of c-Myc and elevation of E2F1, as well as PP2A inactivation.

In this present study, Clark et al. determined the expression of CIP2A in chronic myelocytic leukemia in patients undergoing leukemia progression or treatment failure and explored its potential role as biomarker.

The manuscript is interesting, novel and is well written, clear, and comprehensive. 
The only critical points are as following:

Comments:

  • Did the authors observe that the high diagnostic CIP2A was strongly associated with the presence of a particular CML mutation?
  • Did the authors perform any multivariate analysis of known predictors of outcome like age, cytogenetics, white count, together with CIP2A and related proteins (PP2AY307, SET, SETBP1, c-Myc, AKTS473, STAT5) to derive a prognostic model for survival?
  • Are high levels of AKT phosphorylation (which is inversely correlated with PP2A activity) associated with poor survival in all patients caused by high level of PP2A inhibitors, either CIP2A or SETBP1? If yes, AKT phosphorylation is the biomarker of PP2A inhibition by either CIP2A or SETBP1.
  • A novel CIP2A variant, NOCIVA is been identified and high NOCIVA expression is a marker of a poor clinical outcome. Did the authors considered and checked this variant?
  • The use of many abbreviations, making it difficult to read. I recommend including a list of all abbreviations used in the text and paying attention to write the full names of the acronyms reported in the text.

Round 2

Reviewer 1 Report

The manuscript is well corrected.